# Thermally Deposited Sb_2_Se_3_/CdS-Based Solar Cell: Experimental and Theoretical Analysis

**DOI:** 10.3390/nano13061135

**Published:** 2023-03-22

**Authors:** Raman Kumari, Chandan Yadav, Rahul Kumar, Kamlesh Kumar Maurya, Vidya Nand Singh

**Affiliations:** 1Academy of Scientific and Innovative Research (AcSIR), Ghaziabad 201002, India; 2CSIR-National Physical Laboratory, Dr. K. S. Krishnan Marg, New Delhi 110012, India

**Keywords:** Sb_2_Se_3_ thin film, I–V, thickness, SCAPS-1D

## Abstract

As a promising solar absorber material, antimony selenide (Sb_2_Se_3_) has gained popularity. However, a lack of knowledge regarding material and device physics has slowed the rapid growth of Sb_2_Se_3_-based devices. This study compares the experimental and computational analysis of the photovoltaic performance of Sb_2_Se_3_-/CdS-based solar cells. We construct a specific device that may be produced in any lab using the thermal evaporation technique. Experimentally, efficiency is improved from 0.96% to 1.36% by varying the absorber’s thickness. Experimental information on Sb_2_Se_3_, such as the band gap and thickness, is used in the simulation to check the performance of the device after the optimization of various other parameters, including the series and shunt resistance, and a theoretical maximum efficiency of 4.42% is achieved. Further, the device’s efficiency is improved to 11.27% by optimizing the various parameters of the active layer. It thus is demonstrated that the band gap and thickness of active layers strongly affect the overall performance of a photovoltaic device.

## 1. Introduction

Thin-film-based absorber layers are an intriguing option for the development of low-cost, large-scale photovoltaic systems and have already attracted the attention of many researchers. In thin-film-based photovoltaic cells, active layer materials which are several micrometers thick are used compared to silicon-based solar cells. These materials have a higher absorption coefficient than crystalline ones. As a result, only a very thin absorber layer is needed for sunlight absorption. Nowadays, researchers are moving towards chalcogenide thin-film-based devices, which are reaching their commercial stage due to their outstanding performance and higher efficiency. Metal chalcogenides such as cadmium telluride (CdTe) [1], copper gallium diselenide (CuGaSe_2_) [2], antimony selenide (Sb_2_Se_3_) [3], etc., provide a range of optical bandgaps, which are beneficial for use in many optical and optoelectronic applications. Sb_2_Se_3_ is a promising absorber material, having a direct bandgap of 1.2 eV [4,5,6] and a higher absorption coefficient of ~10^5^ cm^−1^ [7]. These are very favourable for use in optoelectronic applications.

The fabrication of Sb_2_Se_3_ material has been carried out using a variety of deposition processes, such as chemical vapour deposition (CVD) [8], atomic layer deposition (ALD) [9], sputtering [10], spray pyrolysis [11], pulsed laser ablation (PLD) [12], thermal vapour deposition [13], etc. A Sb_2_Se_3_ thin-film-based solar cell with a 6.06% efficiency was fabricated with a two-step sputtering and post-selenization technique [14]. Another Sb_2_Se_3_ thin-film-based solar cell achieved an efficiency of 7.4%. In preparing this, the absorber layer was deposited using the vapour transport method and post-selenized [15]. In other research, after selenization, the well-crystallized Sb_2_Se_3_ thin films were produced with the correct orientation and large crystal grains and an efficiency of 4.86% was attained [16]. Most earlier studies on Sb_2_Se_3_-based solar cells only used solution processing methods. Although these procedures can be inexpensive, they frequently contaminate the film or use toxic solvents. Thermal vapour deposition is a vacuum-based deposition technique. It is used widely in the commercial deposition of thin-film-based structures because it is easy to use and reliable. Sb_2_Se_3_ has a high vapour pressure and low melting point, promoting simple thermal evaporation over magnetron sputtering. The deposited films made using the thermal deposition method are usually categorized as amorphous, and their crystallinity can develop through heat treatment. Until now, no report on a Sb_2_Se_3_ thin-film solar structure has been entirely fabricated using a vacuum-based process.

CdS is one of the promising buffer layers in photovoltaic cells [17,18,19]. CdS films must have good transparency and be neither thin nor thick to prevent short circuits or light absorption in the buffer layer [20]. The higher optical band gap of a hexagonal buffer layer makes it preferable over a cubic layer for use in high-efficiency solar cells. The p-Sb_2_Se_3_-/n-CdS-based solar cells, where the layers are deposited via thermal vapour deposition and then annealed to produce crystallinity, constitute the primary focus of this research. Various characterizations have been analyzed in order to study the properties of active layers. XRD, UV–Vis, and Raman characterizations [21,22,23,24] are analyzed to ensure the structure, bandgap, and phase formation of the functional layers, respectively. The material’s bandgap and thickness significantly influence the performance of a solar cell. Structural analyses supported the appearance of the orthorhombic and hexagonal phases for Sb_2_Se_3_ and CdS, respectively (XRD and Raman). I–V characteristics are plotted after the fabrication of the device.

Since Sb_2_Se_3_ is still in its early stages of development, simulations are crucial to saving the researchers’ time and energy while optimizing the properties of the materials and devices. The parameters of p-Sb_2_Se_3_ obtained from experimental study, such as thickness, bandgap, and absorbance, are used to analyze the efficiency of a Sb_2_Se_3_-/CdS-based photovoltaic cell. The problems preventing the development of a more efficient cell must be resolved and optimized using proper numerical study. This work is based on the experimental and numerical analysis of Sb_2_Se_3_-/CdS-based photovoltaic cells. The SCAPS-1D is used for the device’s simulation study [25]. The I–V curve provides the solar cell parameters, allowing researchers to study a photovoltaic cell’s electrical behaviour. 

## 2. Experimental Study of the Proposed Device Structure

### 2.1. Material Preparation and Characterization Techniques

Highly pure Sb and Se were placed in a silica tube in a stoichiometry ratio of 2:3. The tube was evacuated at a pressure of ~10^−5^ torr. The silica tube was placed in a furnace and heated to 800 °C. After this, the temperature was maintained for 48 h, as shown in Figure 1. The furnace was turned down, and the tube was left to cool to room temperature. The collected ingots were crushed, and a black powder was obtained. Different characterization techniques can be used to analyze the various properties of the materials [26]. The active layer’s structures were ensured by XRD (Cu-Kα, 0.1542 nm). The UV–Vis spectrometer (AvaLight-DH-S-BAL) was used to investigate the optical properties of Sb_2_Se_3_ in the 200–1000 nm wavelength range. With the help of a stylus profilometer, the film’s thickness was measured (Model XP-200). Raman spectroscopy was used to analyze the phase of thin films (Triple Jobin Yvon T 6400).

### 2.2. Device Fabrication via Thermal Deposition

The structure of the solar device is ITO/CdS/Sb_2_Se_3_/Ag, as shown in Figure 2a, and the device was fabricated in a superstrate configuration [27]. From the experimental point of view, we successfully fabricated the device based on the different thicknesses of the absorber layer using the thermal vapour deposition method shown in Figure 2b. The deposition chamber was pressurized to a pressure of 10^−6^ torr before evaporation. As the applied current was continuously increased, the powder in the boat melted and evaporated. First, the CdS film was deposited onto ITO glass with a thickness of 65 nm. Then, two different thicknesses (372 nm and 640 nm) of Sb_2_Se_3_ were deposited onto CdS. Lastly, the silver electrode (Ag) was deposited. The absorber and buffer layers of the device were annealed at 200 °C and 300 °C for 30 and 20 min, respectively, after deposition. The active area of solar cells was ~1 cm^2^, and there was no anti-reflective coating.

### 2.3. Characterization Results

The XRD pattern for the Sb_2_Se_3_-based thin films with two different thicknesses (372 nm and 640 nm) is shown in Figure 3a. There is a sharp peak in the thin film XRD data for both thicknesses (221). When the XRD pattern was compared to Sb_2_Se_3_′s standard JCPDS values, it was concluded that the structure was orthorhombic (JCPDS no.15-0861). The XRD pattern demonstrated sharp peaks for the thin film with a 640 nm thickness. In XRD, sharp peaks analysis indicated that the films were crystalline [28]. The intensity of the peaks likewise increased as the thickness rose. Similarly, in Figure 3b, the XRD for CdS is compared with the standard that of JCPDS no. 89-0019, and the peaks confirm the existence of a hexagonal structure.

Figure 3c depicts the variation in (αhυ)^2^ with photon energy (hυ) for the direct bandgap of samples with thicknesses 372 nm and the 640 nm, respectively. The absorption spectra are shown in Figure 3c inset, where light absorption in the UV–visible region is enhanced as the Sb_2_Se_3_ thin film’s thickness increases. This work’s data agree with those from earlier investigations into Sb_2_Se_3_ thin films, which were performed using different techniques [29]. In earlier reports, the thin film’s absorption coefficient ‘α’ was determined to be a function of wavelength variation in bandgap resulting from various deposition techniques and synthesis conditions [30]. Raman spectroscopy, a quick and, ideally, non-destructive method, can be used to characterize Sb_2_Se_3_, CdS and their impurity phases. Raman spectroscopy is based on the inelastic scattering that results from the interaction of monochromatic light with the material. The shift in the frequency of the emitted photons is a property that characterizes numerous frequency modes, including rotational, vibrational, and other low-frequency transitions in the molecules. Raman mode symmetry determines the nature of a material’s vibrations. The peaks firmly match those that have already been recorded [31]. The results are shown in Figure 3d. Both films made of Sb_2_Se_3_ have a vibrational mode at 189 cm^−1^. For a 640 nm thickness, a new peak (253 cm^−1^) is observed. The Se-Se bond’s stretching vibrations cause the development of a peak at 253 cm^−1^. The vibrational mode is broader for the film at a lower thickness.

In contrast, the band becomes sharper as thickness increases. The band widens in cases of stress or a structural flaw, suggesting poor lattice structural quality. The intensity of both modes increases with the film thickness to support the increased crystalline nature of the film. Similarly, the Raman spectra of thermally deposited CdS films are shown in Figure 3e, where two peaks which are representative of the first and second longitudinal optical phonon modes, have been observed. The 1LO represents the fundamental and overtone modes (301 cm^−1^) and 2LO (~580 cm^−1^) peaks, which were strong and weak, respectively, and almost close to the literature value. Due to the phonon confinement effect, the location of the 2LO mode in the CdS thin films shifted slightly [20,32]. Figure 3f,g depict the field-emission scanning electron microscopy (FE-SEM) and energy-dispersive X-ray (EDAX) analysis for Sb_2_Se_3_ thin film, which is used to ensure the morphology and chemical composition, respectively. The surface of the thin films was uniform and good.

Similarly, in Figure 3h, FE-SEM of CdS is shown and a surface with small and uniform grains that are free of pinholes is found. These characteristics are in good agreement with the film’s high transparency. The Fe-SEM image reveals that the grains were about 15 nm in size.

### 2.4. Experimental Results

#### Current–Voltage Characteristics

The I–V measurement for the solar cell was carried out under standard test conditions (100 mWcm^−2^ light illumination). The I–V curves for various absorber layer thicknesses and corresponding device performance parameters are shown in Figure 4. Power output and cell fill factor have been computed from the illuminated I–V characteristics, which measure the short-circuit current and open-circuit voltage.

The performance of solar cells is often described using the fill factor (FF) and efficiency (η), as shown in Equations (1) and (2)
(1)$$FF=\frac{V_{MP}\times J_{MP}}{V_{OC}\times J_{SC}}$$
(2)$$\eta =\frac{P_{max}}{P_{in}}$$Here, P_max_ is maximum power, P_in_ is incident power, Voc is open-circuit voltage, and Jsc is short-circuited current.

For the 372 nm absorber layer’s thickness, η is achieved as 0.96% (Voc = 0.42 V, Jsc = 5.9 mA/cm^2^, FF = 39%, Rs = 16.7 Ω cm^2^, and Rsh = 342 Ω cm^2^) and for 640 nm, η is achieved as 1.36% (Voc = 0.46 V, Jsc = 6.9 mA/cm^2^, FF = 48%, Rs = 12.5 Ω cm^2^, and Rsh = 386 Ω cm^2^). The results show that the device’s performance improved for the large thickness of the absorber layer.

## 3. Numerical Study of the Proposed Device Structure

### 3.1. Simulation Parameter and Working Conditions

In order to fabricate a complete photovoltaic device, it is necessary to simulate the performance before the deposition of experimentally constructed layers. The Sb_2_Se_3_-based structure is modelled and simulated with the help of SCAPS-1D software. ITO/n-CdS/p-Sb_2_Se_3_/Ag is the proposed design of the device employed in this work. ITO and Ag are used as the front and back contacts, respectively. In order to check the device’s efficiency, the p-Sb_2_Se_3_ layer’s thickness is varied based on the experimental data. When simulating solar structures numerically, bandgap variation is also taken into account. The other parameters for Sb_2_Se_3_, CdS, and ITO are chosen from the previous research [33,34]. At its front contact, the solar cell receives 1 (kW/m^2^) power from the AM 1.5G solar light spectrum. Table 1 lists the simulation parameters for a solar structure based on Sb_2_Se_3_, CdS, and ITO materials at 300 K.

### 3.2. Numerical Results

#### 3.2.1. I–V and QE Characteristic

Figure 5a depicts the I–V characteristics curve of the solar cell of the given Ag-/Sb_2_Se_3_-/CdS-/ITO-based structure for two different thicknesses (372 nm and 640 nm) of p-Sb_2_Se_3_ layers, which were achieved from the experimental analysis. We can see that the Sb_2_Se_3_ layer, which is 640 nm thick, exhibits the highest efficiency from those displayed on this I–V characteristics curve. Efficiencies of 2.59% (Voc = 0.68 V, Jsc = 6.98 mA/cm^2^, FF = 53.20%) and 4.42% (Voc = 0.71 V, Jsc = 10.56 mA/cm^2^, FF = 58.76%) were achieved for the 372 nm and 640 nm thicknesses of Sb_2_Se_3_, respectively (Table 2). Figure 5b depicts the external quantum efficiency (EQE) diagram of the Sb_2_Se_3_ thickness-based solar structure. Quantum efficiency indicates how well a solar cell can absorb carriers from photons of a specific energy when they are incident. A thinner absorber causes less photon absorption to occur at longer wavelengths. Fewer photogenerated electron-hole pairs are found inside the absorber layer as a result. A solar cell’s response to various wavelengths is related to quantum efficiency. The device’s absorption was in the 300–900 nm wavelength range. A solar cell with a 640 nm thick absorber layer gives higher QE than the 372 nm thick absorber-based cell. Quantum efficiency drops to zero at larger wavelengths because the light is not absorbed below the bandgaps at longer, lower-energy wavelengths. Both simulated devices show a reduction in quantum efficiency at wavelengths below 350 nm because the ITO substrate primarily absorbs the light.

#### 3.2.2. The Parasitic Factors: Series and Shunt Resistance 

Resistance in a semiconductor’s bulk, connections, contacts, etc., is called it’s series resistance. The shunt resistance comprises lattice flaws and leakage currents via the edge of the photovoltaic cell. Due to the thinness of the active layer materials, these losses happen when some photons pass through it. Figure 6 represents the influence of series (R_S_) and shunt (R_Sh_) resistance on CdS-/Sb_2_Se_3_-based cells’ performances in ranges of 4–20 Ω-cm^2^ and 100–500 Ω-cm^2^, respectively, as the R_s_ and R_sh_ achieved from the experimental study are within these ranges. The solar cells must have low series and high shunt resistances in order to be used in the design and manufacture of high-efficiency PV devices. Here, in the below figures, with an increase in R_S_ and R_Sh,_ the efficiency decreases and increases, respectively. These simulation results show that the R_S_ and R_Sh_ significantly impact a designed solar cell’s performance.

In comparison to the simulation results, the experimental fill factor is lower. The resistive parameters primarily affect the fill factor of a solar device. Though the series and shunt resistances are kept the same, this difference can be due to other losses in the experimental study. These limit the performance of solar cells via phenomena such as optical loss due to non-absorption, thermalization, reflection, transmission, and area loss during the fabrication of a device. Additionally, interface defect density can also affect the fill factor.

#### 3.2.3. Improvement in Device Performance

Optical parameters like bandgap and absorption coefficient should allow the solar spectrum to increase the device’s performance. Numerous energetic photons will traverse the material if the bandgap is too broad, but they will not form an electron-hole (e–h) pair, and in the case of the small band gap, more energy will be wasted as heat. Therefore, materials are preferred whose band gap is specific. The material’s absorption coefficient should also be high because more absorption will occur in that condition, and electrons will excite in the conduction band. The device’s performance is also significantly influenced by the active layer thickness. Thus, a highly efficient solar cell can be manufactured with a better knowledge of the above factors. A few factors must be investigated in order to improve the device’s performance, which places a substantial load on experimental research. Without fabricating the device, numerical analysis is the better and easiest approach to comprehending the physical mechanisms of a photovoltaic cell. In order to overcome and optimize the difficulties in the direction of a more effective solar cell, proper modeling is necessary. Numerical simulations offer a quick and effective method for identifying the key factors that affect better performance and also produce a valuable and realistic device. The efficiency of the Sb_2_Se_3_/CdS structure, shown in Figure 7. is improved from 4.42 to 11.27% by simply increasing the thickness and band gap of the absorber layer from 640 nm to 2 µm and 1.47 to 1.2 eV, respectively. Table 3 demonstrates the comparison study for various fabrication methods used to fabricate the Sb_2_Se_3_-/CdS-based solar cells.

## 4. Conclusions

All the layers of solar cells were thermally deposited by the evaporation technique. These thermally evaporated CdS/Sb_2_Se_3_ solar cells showed open circuit voltages of 0.42 V and 0.46 V for the two different absorber layer’s thicknesses, 372 nm, and 640 nm, respectively. These compare favourably to the reported values for a similar structure. Furthermore, experimentally, the highest efficiencies of 0.96% and 1.36% are achieved for these thicknesses of the absorber. Experimental work is then compared with theoretical study with the help of simulation, and the results state that the efficiency (2.56% and 4.42%) increases with the increase in absorber layer thickness. Series and shunt resistance, obtained from the experimental results, are also used in the simulation work, and their impact on the device’s efficiency can also be noticed. In the simulation study, by further improving the absorber’s band gap from 1.47 eV to 1.2 eV, a maximum efficiency of 11.27% is obtained. This study shows that a one-step thermal deposition technique and a proper numerical approach can produce a highly efficient solar cell.

## Figures and Tables

**Figure 1 nanomaterials-13-01135-f001:**
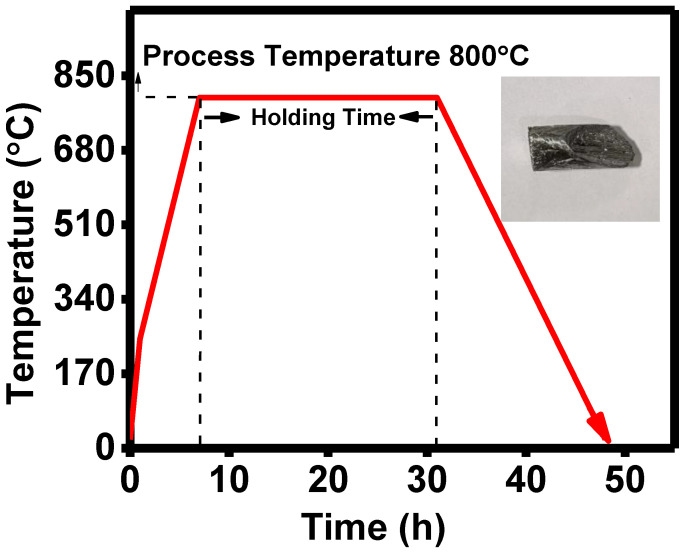
The temperature profile of material inside the vertical tube furnace (inset: single crystal ingot of Sb_2_Se_3_).

**Figure 2 nanomaterials-13-01135-f002:**
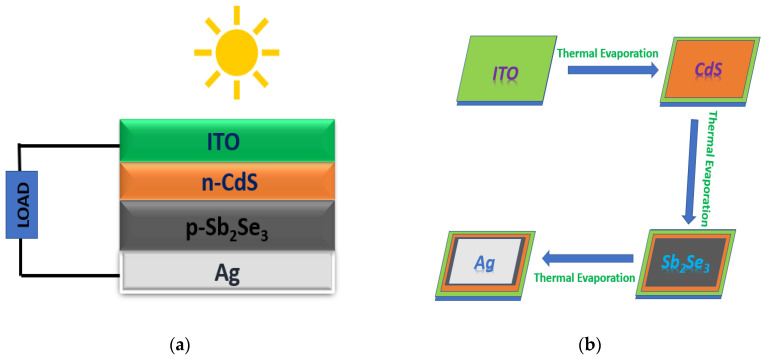
Schematic of (**a**) device structure and (**b**) deposition method.

**Figure 3 nanomaterials-13-01135-f003:**
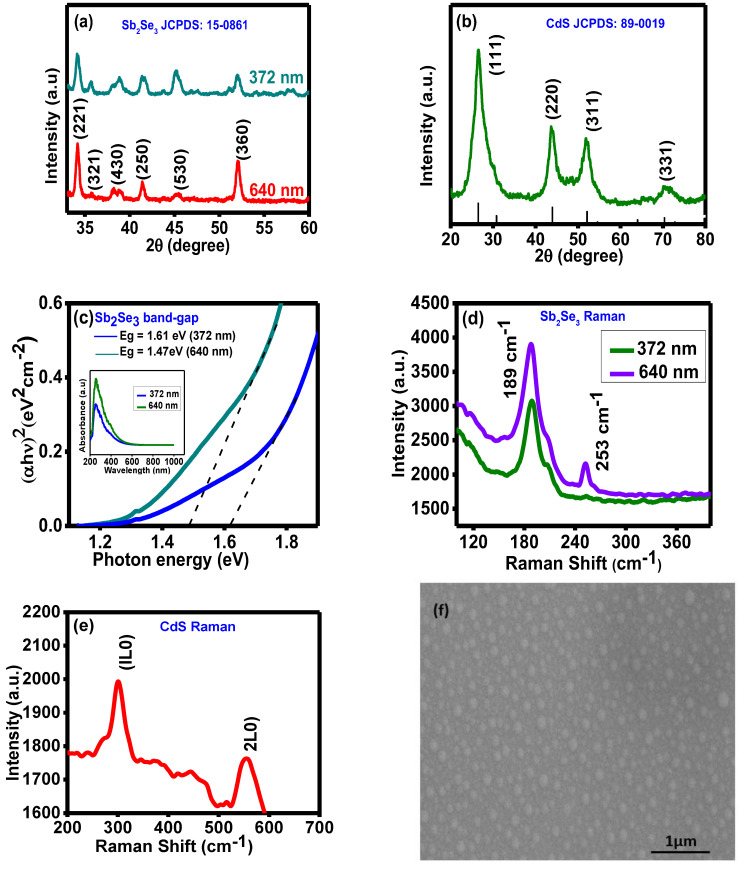
(**a**) XRD analysis for Sb_2_Se_3_ thin films, (**b**) XRD analysis for CdS (inset: absorbance vs. wavelength spectra), and (**c**) band gap of Sb_2_Se_3_ thin films at room temperature. Raman analysis for (**d**) Sb_2_Se_3_ thin films and (**e**) CdS thin films. (**f**) FE-SEM analysis for Sb_2_Se_3_ thin films, (**g**) EDAX spectroscopy for Sb_2_Se_3_ thin films, and (**h**) FE-SEM analysis for CdS thin films.

**Figure 4 nanomaterials-13-01135-f004:**
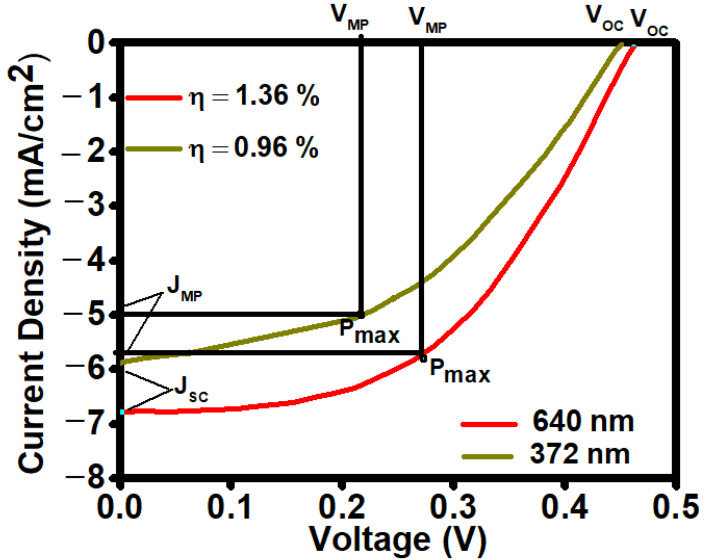
I–V plot for Sb_2_Se_3_/CdS-based structure with different absorber layer thicknesses.

**Figure 5 nanomaterials-13-01135-f005:**
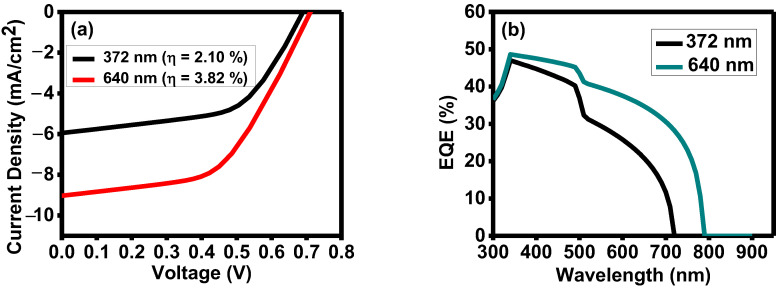
(**a**) I–V and (**b**) QE characteristics for two different thicknesses.

**Figure 6 nanomaterials-13-01135-f006:**
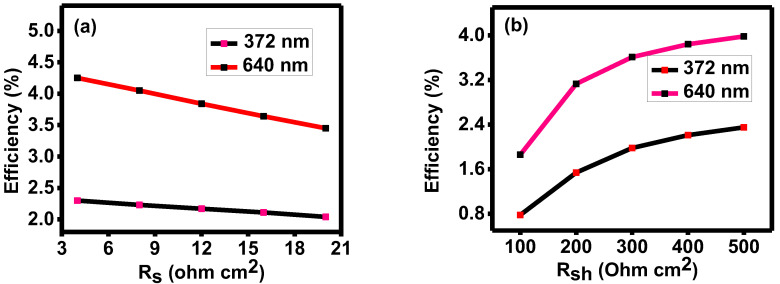
Variation in efficiency with (**a**) series resistance and (**b**) shunt resistance of the device.

**Figure 7 nanomaterials-13-01135-f007:**
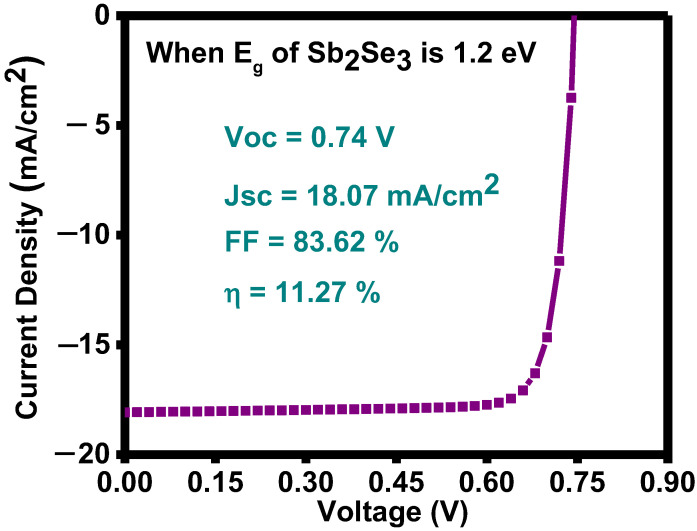
V performance of the optimized solar structure.

**Table 1 nanomaterials-13-01135-t001:** Simulation parameters for Sb_2_Se_3_-/CdS-/ITO-based solar structure.

Parameters	Sb_2_Se_3_	n-CdS	ITO
Thickness	372 nm, 640 nm (Experimental)	65 nm	200 nm
E_g_ (eV)	1.61, 1.47 (Experimental)	2.4	3.72
χ (eV)	4.04	4.2	4.5
ε_r_ (relative)	18	10	9.4
µ_n_ (cm^2^/Vs)	15	100	30
µ_p_ (cm^2^/Vs)	5.1	25	5
Donor density (cm^−3^)	0	10^17^	10^17^
Acceptor density (cm^−3^)	10^18^	0	0

**Table 2 nanomaterials-13-01135-t002:** Comparison of experimental and theoretical results for two different thicknesses of active Sb_2_Se_3_ layer.

Thickness (nm)	Voc (V)	Jsc (mA cm^2^)	FF (%)	η (%)
372 (Exp.)	0.42	5.9	39	0.96
640 (Exp.)	0.46	6.9	43	1.36
372 (Theo.)	0.68	6.98	53.20	2.56
640 (Theo.)	0.71	10.56	58.76	4.42

**Table 3 nanomaterials-13-01135-t003:** Comparison of Sb_2_Se_3_-/CdS-based solar cells fabricated by various methods.

S.no.	Solar Structures	Fabrication Method	Efficiency (%)	References
1.	Mo/Sb_2_Se_3_/CdS/i-ZnO/ITO	E-beam evaporation/CBD	4.34	[35]
2.	FTO/Sb_2_Se_3_/CdS/ZnO/ZnO:Al/Au	Thermal/CBD/sputtering	2.1	[36]
3.	Mo/Sb_2_Se_3_/CdS/i-ZnO/Al-doped ZnO/Ni/Ag	Sputtering/CBD	1.47	[37]
4.	Ag/Sb_2_Se_3_/CdS/ITO	Thermal evaporation	1.36	This work

## Data Availability

Data will be available upon valid request. The raw/processed data required to reproduce these findings cannot be shared as the data also forms part of an ongoing study.

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
