# Peer review of "Thermally Deposited Sb2Se3/CdS-Based Solar Cell: Experimental and Theoretical Analysis"

_nanomaterials, 2023, doi:10.3390/nano13061135_

Round 1

Reviewer 1 Report

THE manuscript is presenting an interesting topic in the investigation of Sb2Se3/CdS based thin film solar cells. the experiment and theory were performed in parallel. the few comments and questions are due before we proceed with this submission:

1- fig. 7 must be drawn in quarter 4 of the framework so u should limit the voltage and current to 0 axis and let this be represented properly as a IV curve without extra current shown above for I>0.

2- in table 2, the gap between experiment and theory of fill factor values is too much. the series resistant is too much on the surface before you deposit the SbSe on contact or on surface of CdS maybe due to dust or moisture.

2-

3-in introduction citation is required to https://doi.org/10.1016/j.spmi.2015.09.023

4- in section 2.4.1 if you cancluate the efficiency by using the (Voc = 0.42 V, Jsc = 5.9 mA/cm2 , FF = 39 %) you should get 9.66% efficiency but authors are reporting 9.1%. this questions the whole manuscript and measurements.

5- i have never seen an anomaly as see near V=Voc of Fig. 4 for 640nm. This is not a roll-over neither a crossover of analysis but seems to be a noise in data measured.

When authors take my comments in and reply to my questions above i can reconsider my decison. do not skip any of above comments/questions.

Author Response

Thank you very much for the comments. We made every effort to respond to each comment. The revised manuscript highlights all the changes and includes all the necessary revisions.

Comment 1: Fig 7 must be drawn in quarter 4 of the framework so you should limit the voltage and current to 0 axis and let this be represented properly as a I-V curve without extra current shown above for I>0

Response: In the revised manuscript, fig 7 has been modified as suggested by the reviewer.

Comment 2: In table 2, the gap between the experiment and theory of fill factor is too much, the series resistance is too much on the surface before you deposit the SbSe on contact or on the surface of CdS may be due to dust or moisture.

Response:  We agree with the reviewer’s comment. In comparison to the simulation, the experimental fill factor is lower. Mainly, the resistive parameters affect the fill factor of a solar device. Though the series and shunt resistances are kept the same, this difference can also be due to other losses in the experimental study, which limit the performance of solar cells such as optical loss that occurs due to non-absorption, thermalization, reflection, transmission, and area loss during the fabrication of device. The interface defect density can also affect the fill factor.

Comment 3: In introduction, citation is required to http://doi.org/10.1016/j.spmi.2015.09.023

Response: In the revised manuscript, the above-said paper has been cited in the introduction section.

Comment 4: In section 2.4.1 if you calculate the efficiency by using the (Voc = 0.42 V, Jsc = 5.9 mA/cm2, FF = 39%) you should get 9.66 % efficiency but authors are reporting 9.1 %, this questions the whole manuscript and measurements.

Response: In the revised manuscript, the value of efficiency has been corrected as per the revised experimentally obtained I-V curve (Fig. 4).

Comment 5: I have never seen an anomaly as seen near V=Voc of fig. 4 for 640 nm. This is not a rollover nor a cross-over of analysis but seems to be a noise in the data measured.

Response: We have rechecked and corrected the I-V curve plot for 640 nm absorber layer thickness and removed the noisy values after 0.46 V.

Reviewer 2 Report

This manuscript is not ready for publication. In the experimental section, there is a lack of some experimental results. The chemical composition of the Sb2Se3 absorber film needs to be analyzed and reported. For the device measurements, it is mandatory to report the external quantum efficiency spectrum. It is also necessary to interpret the existing results in more detail such as the grain size analysis for the Sb2Se3. The section on numerical analysis of the device should be placed before the experimental results. The introduction should mention relevant prior work and these should be compared with the present work in the form of a table at the end of the paper. Equations generally should not be included in the introduction 

There are quite a number of typographic errors in the manuscript. Please revise and improve the English before re-submission. 

Author Response

Comment: This manuscript is not ready for publication. In the experimental section, there is a lack of experimental results. The chemical composition of the Sb2Se3 absorber film needs to be analyzed and reported. For the device measurements, it is mandatory to report the external quantum efficiency spectrum. It is also necessary to interpret the existing results in more detail such as the grain size analysis for the Sb2Se3. The section on the numerical analysis of the device should be placed before the experimental results. The introduction should mention relevant prior work and these should be compared with the present work in the form of a table at the end of the paper. Equations generally should not be included in the introduction. There are quite a number of typographic errors in the manuscript. Please revise and improve the English before re-submission.

Response: Thank you very much for the valuable comments. Morphology and chemical composition’s details for Sb2Se3, with the help of SEM and EDAX have been added in the revised manuscript and are also shown in the below figure. We fabricated the device two months before, and now experimentally, we are unable to plot the quantum efficiency graph for the same results. In the future, we will consider the quantum efficiency plot. In the table (3), the present work is compared with the previous relevant studies based on Sb2Se3/CdS-based thin-film solar structures. We used experimental parameters (series and shunt resistances, thickness, and band gap of absorber layer) in the numerical study, so the experimental section is placed before the numerical section. Equations have been moved to the numerical section. The grammatical and spelling errors have been revised and corrected as suggested by the reviewer.

Fig. 3(f) SEM analysis for Sb2Se3 thin-film

Fig: EDAX spectrum of Sb2Se3 thin-film

Table 3: Comparison of Sb2Se3/CdS based solar cells fabricated by various methods

S.no.

       Solar structures

Fabrication method

Efficiency     (%)

References

1.

Mo/Sb2Se3/CdS/i-ZnO/ITO

E-beam evaporation and CBD

      4.34

    [36]

2.

FTO/Sb2Se3/CdS/ZnO/ZnO:Al/Au

Thermal, CBD and sputtering

      2.1

    [37]

3.

Mo/Sb2Se3/CdS/i-ZnO/Al-doped ZnO/Ni/Ag

Sputtering and CBD

      1.47

    [38]

4.

Ag/Sb2Se3/CdS/ITO

Thermal evaporation

      1.36

This work

Reviewer 3 Report

Review comments for

Manuscript ID: nanomaterials-2278627-peer-review-v1      

Title: Thermally deposited Sb2Se3/CdS-based solar cell: Experimental and theoretical analysis

Mamta Mamta, Raman Kumari, Chandan Yadav, Rahul Kumar, Kamlesh Maurya *, Vidya Nand Singh *

Submitted to: Nanomaterials

Comments:

The submitted manuscript reported thermally deposited Sb2Se3/CdS-based solar cells. Although Sb2Se3/CdS solar cells are reported numerous times in the literature, however, solar cell fabricated entirely based on vacuum thermal deposition is missing. Therefore, reporting this work would be appropriate for publication.

The manuscript is well-written and organized well. However, the novelty and importance of this work, compared to the existing literature are not highlighted enough. Therefore, a few important points are listed below for the improvement of the work. If the authors can clearly demonstrate the difference between this work and other works published in the literature already, then this will be suitable for consideration. The manuscript in its current format should be recommended as a Minor revision.

The issues listed below are critical for the publication and need major consideration to justify all manuscript claims, logical justifications, and clarity.

11)      An introductory figure should be added to the revised manuscript highlighting the absence of Sb2Se3-based solar cells entirely fabricated using a vacuum-based process. And this work focused on this gap. This will justify the novelty and importance of this work.

22)      Any morphological analysis (such as SEM, AFM, OM, etc.) of the Sb2Se3 and CdS thin films should be added to the revised version of the manuscript.

33)      The numbers two and three in Sb2Se3 should be subscript, not superscript (Sb2Se3) as written on page 2, line 61. This should be corrected.

44)      Why there is a large difference between experimental and theoretical solar cell efficiencies? For example see Table 2, for 372 nm Sb2Se3, theoretical η = 2.56%, however, experimentally η was only achieved around 0.91%. The same observation was repeated in the 640 nm Sb2Se3 film. Why is a large difference? Is it because of Rs and Rsh? This is not clearly written in the manuscript. An elaborate discussion should be added to the revised manuscript justifying the large difference between theoretical and experimental values in the description of Table 2.

Author Response

Comment 1: An introductory figure should be added to the revised manuscript highlighting the absence of Sb2Se3-based solar cells entirely fabricated using a vacuum-based process. And this work focused on this gap. This will justify the novelty and importance of this work.

Response: In the revised manuscript, the figure has been added that demonstrates the complete Sb2Se3-based solar structure that is fabricated by a single-step deposition method, i.e., thermal deposition.

Comment 2: Any morphological analysis (such as SEM, AFM, OM, etc.) of the Sb2Se3 and CdS thin films should be added to the revised version of the manuscript.

Response: SEM for Sb2Se3 (fig. f) and CdS (fig. h) has been added to the revised manuscript.

Comment 3: The numbers two and three in Sb2Se3 should be subscript, not superscript (Sb2Se3) as written on page 2, line 61. This should be corrected.

Response: The above said line has been corrected in the revised manuscript.

Comment 4: Why is there a large difference between experimental and theoretical solar cell efficiencies? For example, see Table 2, for 372 nm Sb2Se3, theoretical η = 2.56%, however, experimentally η was only achieved around 0.91%. The same observation was repeated in the 640 nm Sb2Se3 film. Why is a large difference? Is it because of Rs and Rsh? This is not clearly written in the manuscript. An elaborate discussion should be added to the revised manuscript justifying the large difference between theoretical and experimental values in the description of Table 2.

Response: Thank you for the comment. Compared to the simulation, the experimental output is low such as efficiency and fill factor. Mainly, the resistive parameters affect the fill factor of a solar device. Though the series and shunt resistances are kept the same, this difference can also be due to other losses in the experimental study, which limit the performance of solar cell, such as optical loss occurs due to non-absorption, thermalization, reflection, transmission, and area loss during the fabrication of the device. The interface defect density can also affects the fill factor. That is why in a simulation study, higher efficiency is achieved as compared to an experimental study.

Thermalization loss: Thermalization takes place when EPh (energy of the incident photon) > EG (Band gap energy), and it applies to photons with energies higher than the band gap. Heat is produced as a result of the extra energy. This heat raises the temperature of the solar cell, which further raises the reverse saturation current due to an increase in intrinsic carrier concentration and minority carrier diffusion length.

Reflection loss: This occurs due to the blocking of the light by top contact, reflection from the top surface, and reflection from the back contact without proper absorption.

Transmission loss: This is due to the finite thickness of the cell and the effect is enhanced in materials having a low absorption coefficient.

Area Loss: This loss is due to metal grid design or metal electrode coverage.

Collection loss: These losses are due to surface and bulk recombination at metal or semiconductor contact and recombination in the depletion region. These recombination losses mainly affect the open circuit voltage. Impurities, crystalline defects, and incomplete chemical bond on semiconductor acts as traps for photoexcited carriers, and recombination on these traps cause the reduction of photocurrent. The reduction in the concentration of impurities and defects can increase the diffusion length of minority carriers and this can decrease the recombination losses in a solar cell.

Contact Loss: Losses are caused by contacts on a solar cell's front and rear surfaces. The back contact has a metallic coating, and the front contact is made of fine grid lines. One of the key strategies for reducing the power losses for cells is to reduce contact resistance. These losses affect the efficiency of thin film solar cells. Therefore it is necessary to reduce these losses in order to increase the efficiency of the solar cell.

Round 2

Reviewer 1 Report

my comments were properly addressed in the revision. i can accept the paper for publication now.

Reviewer 2 Report

The authors have made the required changes except the lack of experimental EQE data.